# A Spatial Location Representation Method Incorporating Boundary Information

**Hui Jiang [1],\* and Yukun Zhang [2]**

1  School of Intelligent Manufacturing, Huainan Union University, Huainan 232038, China
2  School of Electrical Engineering, Anhui Polytechnic University, Wuhu 241060, China
*  Correspondence: aaa-jhui@163.com

**Abstract:** In response to problems concerning the low autonomous localization accuracy of mobile robots in unknown environments and large cumulative errors due to long time running, a spatial location representation method incorporating boundary information (SLRB) is proposed, inspired by the mammalian spatial cognitive mechanism. In modeling the firing characteristics of boundary cells to environmental boundary information, we construct vector relationships between the mobile robot and environmental boundaries with direction-aware information and distance-aware information. The self-motion information (direction and velocity) is used as the input to the lateral anti-Hebbian network (LAHN) to generate grid cells. In addition, the boundary cell response values are used to update the grid cell distribution law and to suppress the error response of the place cells, thus reducing the localization error of the mobile robot. Meanwhile, when the mobile robot reaches the boundary cell excitation zone, the activated boundary cells are used to correct the accumulated errors that occur due to long running times, which thus improves the localization accuracy of the system. The main contributions of this paper are as follows: 1. We propose a novel method for constructing boundary cell models. 2. An approach is presented that maps the response values of boundary cells to the input layer of LAHN (Location-Adaptive Hierarchical Network), where grid cells are generated through LAHN learning rules, and the distribution pattern of grid cells is adjusted using the response values of boundary cells. 3. We correct the cumulative error caused by long-term operation of place cells through the activation of boundary cells, ensuring that only one place cell responds to the current location at each individual moment, thereby improving the positioning accuracy of the system.

**Keywords:** boundary cells; grid cells; place cells; environmental characterization; brain-inspired computing

## 1. Introduction

Environmental cognition is a fundamental skill for mammalian foraging and survival. Physiological studies have indicated that mammals, when freely moving in unfamiliar environments, are capable of maintaining relative spatial relationships to nests or food through specific cognitive mechanisms. This provides them with positional information for navigation in unfamiliar environments and enables real-time updates based on changes in external environmental cues, thus endowing them with strong perceptual abilities in unknown surroundings [1–4]. However, existing mobile robot technologies fail to utilize distance information between themselves and obstacles or walls to update their current position when facing unexpected obstacles or barriers. Therefore, investigating and replicating the environmental cognition mechanisms observed in mammals holds significant importance in enhancing the environmental cognition capabilities of mobile robots and advancing our understanding of biological environmental cognition [5–7].

In 1971, O'Keefe et al. found, in the rat hippocampus, a cell with a selective firing to spatial locations. This cell undergoes firing activity only when the rat is in a spatially specific environmental location [8]. This cell is called a place cell and its corresponding spatial firing

area is called the place field [9,10]. In 2005, Hafting et al. identified another type of cell in the entorhinal cortex of rats that produces periodic firing to specific regions of space grid cells and whose hexagonal firing fields spread throughout the spatial environment with the movement of the rat [11]. Related studies have shown that when rats move freely in a two-dimensional space, grid cells in the entorhinal cortex undergo repetitive firing behavior at specific locations; furthermore, it was noted that their firing activity is highly stable and, as rats continue to explore the environment, the generated grid cells cover the entire environment and complete the spatial representation of the environment [12–14]. Barry et al. proposed an oscillatory interference (OI) to model the hexagonal firing structure of the grid cell. In the model, the self-motion information (direction and speed) of the mobile robot was used as the input of grid cells to update and maintain the grid field [15]. However, the verification of the model remained in the simulation stage and did not realize effective map construction in the real environment. In [16], the rat simultaneous location and mapping (RatSLAM) model, which was based on a rodent model, was investigated. This model centralizes path integration information and external visual scene information into the pose cell and is able to perform navigation and map construction tasks. However, this model does not incorporate the physiological characteristics of the hippocampal structures in the rat brain, which thus leads to a lack of accuracy and a low stability with respect to this model [17]. To address the problems of the insufficient physiological characteristics of the RatSLAM method, Oliver et al. proposed a grid cell to place cell competitive neural network models in a Hebb learning algorithm, based on the phenomenon of lateral inhibition in the rat hippocampus, which conforms to the physiological characteristics of the hippocampal navigation cells and can realize the information transfer and can also map from grid cells to place cells [18]. Yu Naigong et al. similarly used the Hebb learning algorithm in the work of constructing environmental cognitive maps by imitating the hippocampal cognitive mechanism in the rat brain. They also implemented the environmental cognitive map construction through a real physical platform and obtained better experimental results [19]. O'Keefe et al. found that the size of the place cell firing field changes somewhat when the rat moves to the environmental boundary; to explain this phenomenon in their experiments, O'Keefe et al. predicted the existence of a cell in the rat brain that responds to boundary information with a firing response and is able to use this response to correct for position errors in the position of the place cell [20]. In 2008, researchers discovered a new cell type in the rat internal olfactory cortex that fires when the animal approaches a wall or is separated by other obstacles; this new cell type was accordingly named boundary cells [21,22]. In order to investigate the effect of boundary cells on the distribution and localization accuracy of grid cells, Hardcastle et al. replaced the circular environment with a hexagonal environment that was rich in environmental boundary information. Furthermore, the grid cell distribution was rearranged, and the localization accuracy was improved [23].

Based on this, a spatial location representation method (SLRB) incorporating boundary information was proposed, inspired by the mammalian spatial cognitive mechanism, which obtains the boundary cell response values through the mutual excitation and inhibition of direction-aware information, as well as the distance-aware information between the mobile robot and the environment boundary. This method obtains the boundary cell response values by mutual excitation and through the suppression of direction and distance-sensing information between the mobile robot and the environment boundary. The method then maps the boundary cell response values and the self-motion information of the mobile robot to the input layer of LAHN, and then the output layer of LAHN is mapped to the grid cell response values. The grid cell response values are used as the main input source of the place cells and the response values of the place cells are obtained through the competitive Hebb learning network. At the same time, when the mobile robot runs to the boundary cell excitation zone through the activated boundary cells to correct the location information of the place cells, the mobile robot then learns and remembers the location points in a specific space, as well as constructs a spatial location representation map that can accurately express

the current spatial characteristics. A block diagram of the overall system structure is shown in Figure 1.

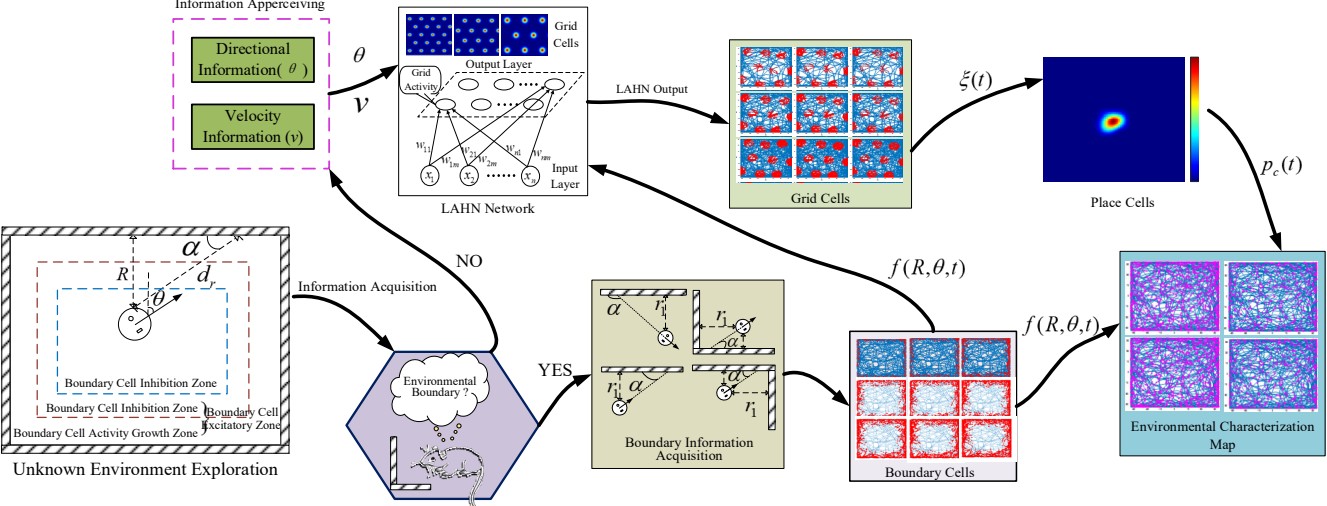

**Figure 1.** Block diagram of the overall system structure.

The main innovations of this paper are as follows:

(1) Inspired by the mammalian spatial cognitive mechanism, a new boundary cell model is proposed to establish boundary cell activity states in multiple scenarios by the mutual excitation and the inhibition of the direction-aware and distance-aware information that is acquired by mobile robots. The boundary cell model proposed in this paper can encode the boundary information in the environment and supplement the lack of environmental boundary perceptual information with path integration.

(2) The physiological phenomena indicate that the environmental boundary information can be used as the supplementary information of grid cells. The method in this paper maps the boundary cell response values to the input layer of LAHN, generates grid cells by LAHN learning rules, and uses the boundary cell response values to correct the grid cell distribution pattern, such that the grid cell firing response and distribution that is activated by the method are more consistent with the physiological characteristics.

(3) According to the problem that the mobile robot runs for a long time in an unknown environment, when the mobile robot reaches the boundary cell excitation zone, the accumulated error caused by the long running time of the position cell is corrected by the activated boundary cells, such that only one place cell responds to the current position at each time in order to improve the location accuracy of the system.

## 2. Spatial Navigation Cell Model

### 2.1. Boundary Cell Modeling

The boundary cells, which mainly exist in the entorhinal cortex, presubiculum, and parasubiculum tract of the rat hippocampus, are spatial navigation cells that respond to boundary information and can reflect the relative positions of rats at different distances and angles from the environmental boundary by encoding boundary information in the environment; these cells can be used to complement path integration information [24–26]. In this paper, the boundary cells are modeled by the mutual excitation and inhibition of direction-aware and distance-aware information of the mobile robot, and the construction process is shown below.

Step 1: The mobile robot explores in an unknown environment (its exploration schematic diagram is shown in Figure 2). The surrounding shaded part is the wall, the circular runner is the mobile robot, and the single black arrow is its movement direction.

The boundary response constant is set to divide the exploration area into the boundary cell activity inhibition zone, the attenuation zone, and the growth zone. The region division rules are as follows:

$$S(t) = \begin{cases} S_{inh} & , \; if \, R(t) > b \\ S_{excar} & , \; if \, b/2 < R(t) \le b \\ S_{excgr} & , \; if \, R(t) \le b/2 \end{cases} \tag{1}$$

where $S(t)$ denotes the region in which the mobile robot is located at the time $t$, $b$ represents the boundary response constant, $S_{excar}$ and $S_{excgr}$ denotes the mobile robot is in the boundary cell inhibition zone, the attenuation zone, and the growth zone, respectively. Furthermore, the boundary cell activity attenuation zone and growth zone are subsets of the boundary cell excitation zone.

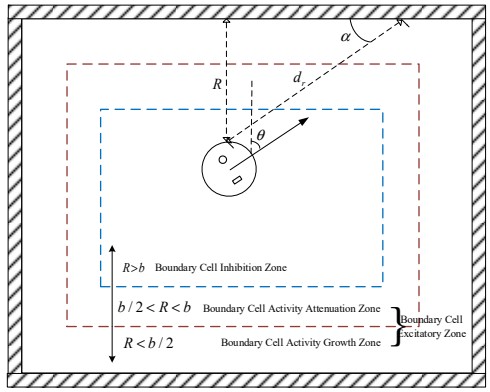

**Figure 2.** Schematic diagram of the mobile robot exploration in the environment.

Step 2: The mobile robot scenes are divided into six scenarios in the environment, as shown in Figure 3. Scenarios A and C depict the mobile robot moving towards and away from a wall, respectively. Scenarios B and D represent movements away from and towards a corner, respectively. Scenarios E and F illustrate movements away from and towards a curved wall, respectively. Where the shaded parts are walls and where the mobile robot updates the perceptual information during its movement:

$$\theta(t+1) = \theta(t) + \theta_s(t)T \tag{2}$$

$$[R(t), d_r(t)] = [\min(r_1(t), r_2(t) \cdots r_n(t)), \min(d_r(t), 50)] \tag{3}$$

$$\alpha = \gamma \arctan \frac{R(t)}{d_r(t)} - (\gamma - 1)(\pi - \arctan \frac{R(t)}{d_r(t)}) \tag{4}$$

where $\theta$ is the direction-aware information of the mobile robot, $\theta_s$ is the angular velocity-aware information of the mobile robot, and $T$ is the sampling period. In this paper, it is set as 0.01 s, which represents the data collected by the mobile robot updated every 0.01 s, $R$ is the distance-aware information of the mobile robot to the nearest environmental boundary, $d_r$ is the distance-aware information between the current position and the environmental boundary directly in front of the mobile robot, $r$ is the vertical distance between the mobile robot and the environmental boundary, and $n$ is the number of environmental boundaries currently detected by the mobile robot. $\alpha$ is the angle information between the mobile robot and the nearest surrounding environmental boundary, $\gamma$ is the regulatory factor of the angle information, and $\gamma$ takes the value of 1 only when the robot's direction of motion is perpendicular to the wall—otherwise it takes the value of 0.

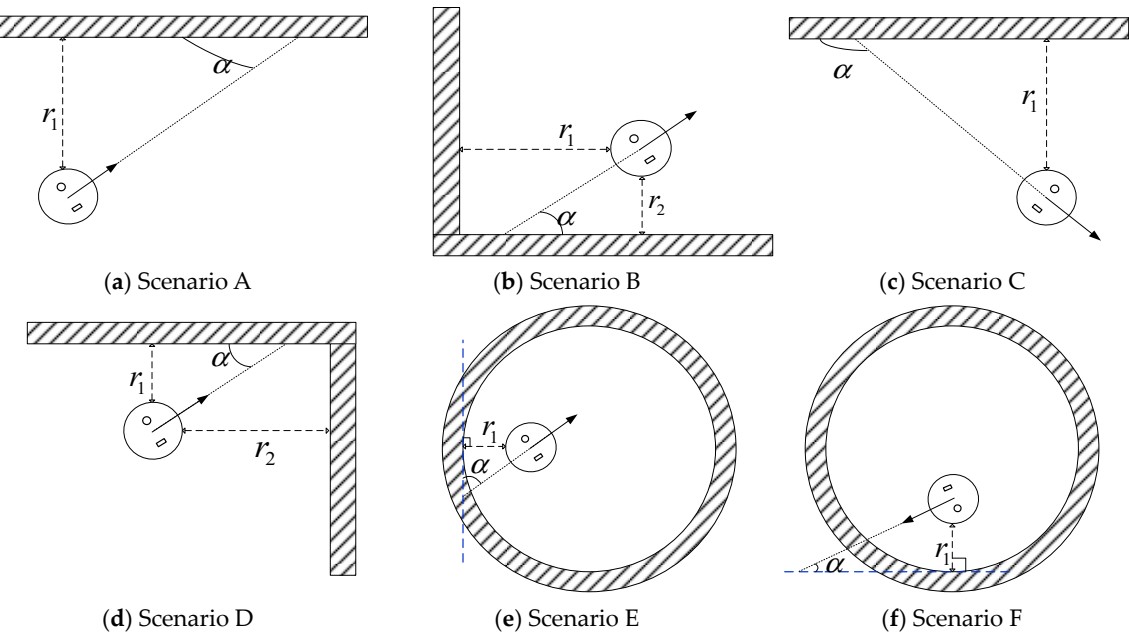

**Figure 3.** Diagram of the six scenarios of mobile robots in the environment.

When the mobile robot moves away from the wall, the boundary cells go through two processes, the activity growth zone and the attenuation zone. When they then reach the boundary cell inhibition zone their activity attenuates to zero. The rules for updating the activity state of boundary cell neurons are as follows:

$$h(t+1) = \begin{cases} h(t) + \tau\frac{b-d_r(t)}{b}, & if\ S(t) = S_{excgr} \\ 0, & if\ S(t) = S_{inh} \\ \left[h(t) - \tau\frac{b-d_r(t)}{b}\right]_+, & if\ S(t) = S_{excar} \end{cases} \tag{5}$$

where $h(t)$ is the current time boundary cell activity value, $h(t+1)$ is the next time boundary cell activity value, and $\tau$ is the activity factor (which is used to regulate the rate of change in boundary cell neuron activity and the value of 0.8 is taken for this factor in this paper). $[\bullet]_+$ indicates that the output value is non-negative.

Step 3: When the mobile robot runs into the boundary cell excitation zone, the boundary cell distance excitation value $b_{border}(t)$ is updated with the distance-aware information between the current position of the mobile robot and the environmental boundary.

$$b_{border}(t) = \exp\left(-\frac{h(t)(R(t) - d_r(t))^2}{2\sigma_{rad}^2 (d_r(t))}\right) \tag{6}$$

where $\sigma_{rad}(\bullet)$ is the boundary cell distance sensitivity function and the relationship between the sensitivity of the boundary cell to the environmental boundary. Moreover, the vertical distance from the mobile robot to the environmental boundary is expressed as per the following:

$$\sigma_{rad}(d_r(t)) = \delta_0 * (d_r(t)/\beta + 1) \tag{7}$$

where $\delta_0$ is the boundary cell sensitivity enhancement constant, which is adapted to the complexity of the environmental boundary and which is set to 1.2 in this paper. $\beta$ is the boundary cell distance perception correction factor, which avoids the boundary cell response value being too large due to the small distance information that is detected in the small environmental scenarios and which is taken as 0.8 in this paper.

Subsequently, the boundary cell angular excitation value $r_{border}(t)$ is updated based on the angular-aware information between the mobile robot and environmental boundary:

$$r_{border}(t) = \exp\left(-\frac{h(t)(\theta(t)-\alpha(t))^2}{2\sigma_{ang}^2}\right) \tag{8}$$

where $\sigma_{ang}$ is the adjustment parameter for the angular excitation value of the boundary cell, which is used to regulate the effect of angle on the excitation value of the border cell, and is taken as 0.5 in this paper.

Step 4: The boundary cell response value is updated by the boundary cell distance excitation value and angular excitation value, and the boundary cell fire response $f(R, \theta, t)$ in the spatial environment is shown as per the following:

$$f(R, \theta, t) = \frac{b_{border}(t) \times r_{border}(t)}{\sqrt{2\pi\sigma_{rad}^2\,(d_r(t))}\sqrt{2\pi\sigma_{ang}^2}} \tag{9}$$

The rats explored freely in the experimental environment and the strain response was different from the environmental scene information, which were collected at different times. The six different boundary cell activation response maps in Figure 4 correspond one by one to the six different experimental scenarios in Figure 3, with the pentagram position as the starting point in Figure 4. As shown in Figure 4a, during the process of a rat running towards a wall, it gradually transitions from the boundary cell inhibition zone to the boundary cell activation zone, resulting in an increase in the number of activated boundary cells. Particularly when entering the boundary cell growth zone, the number of boundary cells rapidly rises. In the boundary cell decay zone, the number of boundary cells decreases accordingly. When reaching the boundary cell inhibition zone, the boundary cells will not be activated. In Figure 4b, the rat is currently moving away from the environmental boundary and the boundary cell activation frequency thus gradually decreases (i.e., the number of boundary cell activation increases slowly). The current movement posture of the rat in Figure 4c corresponds to Figure 3d, thereby showing a tendency to move away from the environmental corners and showing a gradual decrease in the boundary cell activation frequency. The transient movement posture of the rat shown in Figure 4d is similar to that in Figure 4c, but unlike Figure 4c, the rat is away from the corner of the environment at this time, and the binding force of the environment unilaterally on the movement of the rat decreases rapidly as the distance of the rat away from the boundary increases; as such, the number of boundary cell activations temporarily enters a low-rate growth phase. Similarly, the same is shown in Figure 4e,f, which show the boundary cell activation response maps of the rats in different scenarios, where the response maps are acquired by movement in a circular experimental environment, thereby corresponding to rats that are near and far from the environmental boundary in Figure 3e,f, respectively.

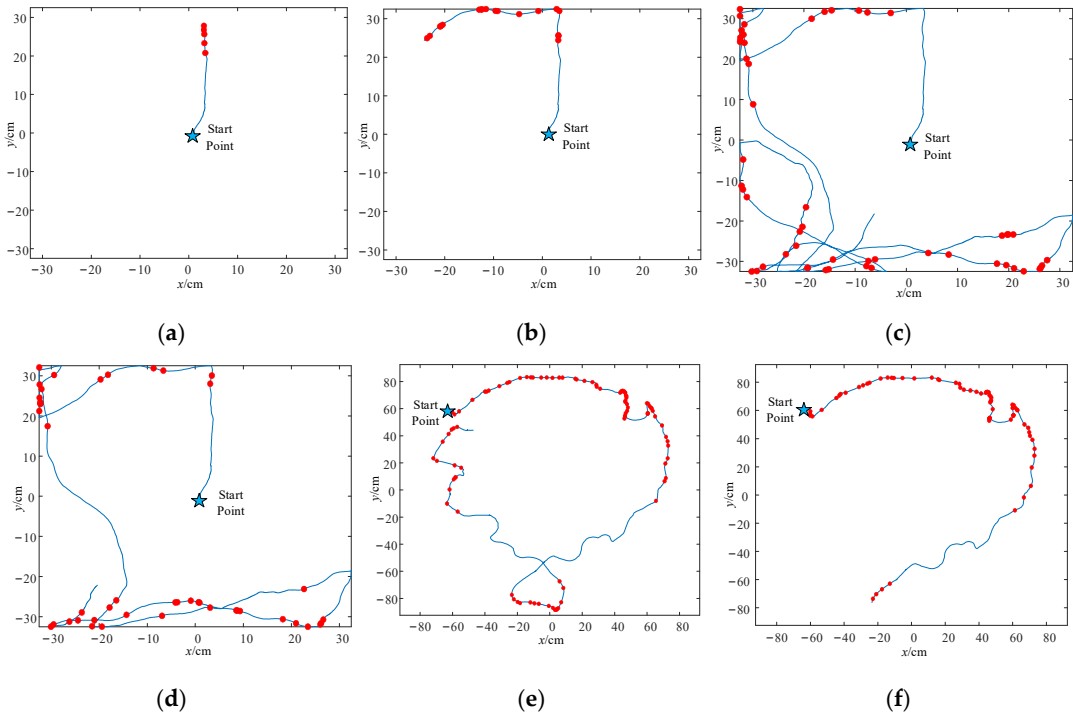

**Figure 4.** Activated boundary cells in different scenarios. (**a**) Activated boundary cells under scenario A. (**b**) Activated boundary cells under scenario B. (**c**) Activated boundary cells under scenario C. (**d**) Activated boundary cells under scenario D. (**e**) Activated boundary cells under scenario E. (**f**) Activated boundary cells under scenario F.

### 2.2. Grid Cell Update Model Based on Boundary Information

LAHN is designed as an unsupervised neural network that can obtain the best features from the input information [27]. The superiority of this network is also reflected in the fact that when the input information is limited by the external environment, the network itself can still update the output in real time by adjusting the lateral connections to adapt to the environmental changes [28]. When considering the influence of environmental boundary information on the distribution pattern of grid cells during the movement of the rat, LAHN is introduced in this paper to model the grid cell update mechanism. The self-motion information and boundary cell response values acquired by the encoder are mapped to the input layer of LAHN during the exploration of the environment by the mobile robot, while the excitation level and inhibition level of the grid cells are updated in real time with the movement of the mobile robot. The update rules of this are as follows:

$$\frac{d\left(\chi_j^{inh}\right)}{dt} = -\chi_j^{exc}\left[f(R,\theta) + vw\begin{bmatrix}\cos(\theta_s)\\\sin(\theta_s)\end{bmatrix}\right] + \chi_j^{inh}\left[1 - \left(\chi_j^{inh\,2} + \chi_j^{exc\,2}\right)\right] \qquad (10)$$

$$\frac{d\left(\chi_j^{exc}\right)}{dt} = \chi_j^{inh}\left[f(R,\theta) + vw\begin{bmatrix}\cos(\theta_s)\\\sin(\theta_s)\end{bmatrix}\right] + \chi_j^{exc}\left[1 - \left(\chi_j^{inh\,2} + \chi_j^{exc\,2}\right)\right] \qquad (11)$$

where $\theta_s$ is the angular velocity-aware information and $W$ is the self-organizing mapping input weight matrix, $\theta$ represents the directional sensory information of the mobile robot. $v$ is the current velocity of the mobile robot, where the value of $\chi_j^{inh}$ is a negative number indicating the inhibition level of the *j*-th grid cell. The value of $\chi_j^{exc}$ is a positive number indicating the *j*-th grid cell excitation level and $f(R,\theta)$ is the boundary cell response value.

LAHN uses a bipolar activation function and the dependent variable of the activation function takes values from −1 to 1, which is when the input and output of the activation

function have the same sign and where the network connection weight is increased, otherwise the network connection weight is instead decreased. The network output value is the grid cell response value and the LAHN output value is shown in Equation (12):

$$\xi_i(t) = \sum_{j=1}^{m} q_{ij}[\chi_j^{exc}(t) + \chi_j^{inh}(t)] + \sum_{k=1}^{n} w_{ik}\xi_k(t-1) \tag{12}$$

where $q_{ij}$ is the forward channel weight, $w_{ik}$ is the lateral channel weight, $\xi_k(t-1)$ is the grid cell response value at the previous time, $m$ is the total number of neurons in the LAHN layer, and $n$ is the number of grid cells.

The grid cell distribution maps and their corresponding grid cell firing rate maps were obtained by exploring in the trilateral, pentagonal, and nine-sided environments, respectively, as shown in Figure 5a,c. The control analysis shows that the grid cell clusters converge with the highest firing rate in the center, decreasing layer by layer toward the periphery. In this paper, we introduced the grid cell scoring mechanism, which is shown in Equations (13) and (14) [29–31]. This mechanism was constructed to score the distribution and activity of the grid cells that are obtained in the three different environments. This mechanism also allowed us to generate a grid cell score table, which is shown in Table 1. The trends of the grid cell scores in the three different environments showed that the grid cell scores gradually increased and eventually stabilized as the exploration time increased. This pattern of data change is due to the positive effect of the boundary cells that was activated by the rats visiting the boundary of the environment, which thus updated the grid cell distribution. The reason for the higher grid cell scores, which were obtained by exploring the nine-sided environment rather than the pentagonal and trilateral ones, is explained by the fact that the nine-sided environment provides richer boundary information.

$$r(\tau_x, \tau_y) = \frac{n\sum\limits_{x,y} \lambda(x,y)\lambda(x - \tau_x, y - \tau_y) - \sum\limits_{x,y} \lambda(x,y)\sum\limits_{x,y} \lambda(x - \tau_x, y - \tau_y)}{\sqrt{\left[M\sum\limits_{x,y} \lambda(x,y)^2 - \left[\sum\limits_{x,y} \lambda(x,y)\right]^2\right]\left[n\sum\limits_{x,y} \lambda(x - \tau_x, y - \tau_y)^2 - \left[\lambda(x - \tau_x, y - \tau_y)\right]^2\right]}} \tag{13}$$

$$HGS = \min\left[\text{cor}\left(r, r^{60°}\right), \text{cor}\left(r, r^{120°}\right)\right] - \max\left[\text{cor}\left(r, r^{30°}\right), \text{cor}\left(r, r^{90°}\right), \text{cor}\left(r, r^{150°}\right)\right] \tag{14}$$

where $n$ is the number of grid cells, *HGS* is the fraction of grid cells, $r$ is the auto-correlation plot of grid cells, and $\lambda(x,y)$ is the firing rate of grid cells at position. Furthermore, $\tau_x$ and $\tau_y$ are the spatial lag coordinates corresponding to the $x$ and $y$ coordinates, and $r^\beta$ is the auto-correlation plot rotated by $\beta$ degrees. $\text{cor}(r, r^\beta)$ is the correlation score of the auto-correlation plot $r$ and the correlation score of the two plots after rotating the auto-correlation plot by $\beta$ degrees.

**Table 1.** Grid cell scores in different geometric environments.

| Time (s) | 30 | 60 | 90 | 120 | 150 | 180 | 210 | 240 | 270 | 300 | 330 | 360 |
|---|---|---|---|---|---|---|---|---|---|---|---|---|
| Trilateral environment | 0.25 | 0.32 | 0.38 | 0.45 | 0.55 | 0.56 | 0.68 | 0.69 | 0.74 | 0.73 | 0.74 | 0.74 |
| Pentagonal environment | 0.24 | 0.35 | 0.41 | 0.51 | 0.62 | 0.68 | 0.72 | 0.75 | 0.78 | 0.78 | 0.78 | 0.78 |
| Nine-sided environment | 0.20 | 0.34 | 0.40 | 0.49 | 0.58 | 0.71 | 0.80 | 0.84 | 0.86 | 0.84 | 0.85 | 0.85 |

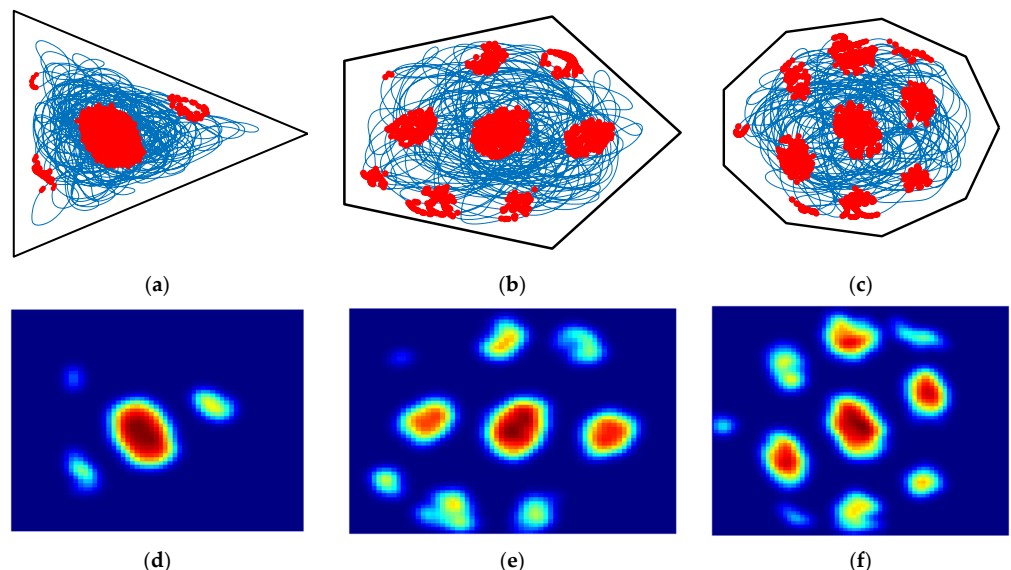

**Figure 5.** Cell distribution and firing rate in different geometric environments. (**a**) Cell distribution in a trigonal environment. (**b**) Cell distribution in a pentagonal environment. (**c**) Cell distribution in a nine-sided environment. (**d**) Cell firing rate in a trilateral environment. (**e**) Cell firing rate in a pentagonal environment. (**f**) Cell firing rate in a nine-sided environment.

## 3. Spatial Location Representation Map Construction

Based on the physiological properties of each the navigation cells mentioned above, it is known that boundary cells can fire specifically in response to the perception of the environmental boundary by the rat, i.e., the vectorial relationship between the rat and the boundary. Furthermore, the closer the rat is to the environmental boundary, the greater the value of the boundary cell firing response [32]. Grid cells are considered as a coordinate system for characterizing the environment due to their specific spatial metric properties, and when multiple grid cells fire aggregately the current position of the rat can be estimated [33,34]. A process schematic diagram of constructing a spatial location representation map, using the specific firing responses of these navigation cells and their mapping relationships with each other, is shown in Figure 6. The specific map construction steps are as follows:

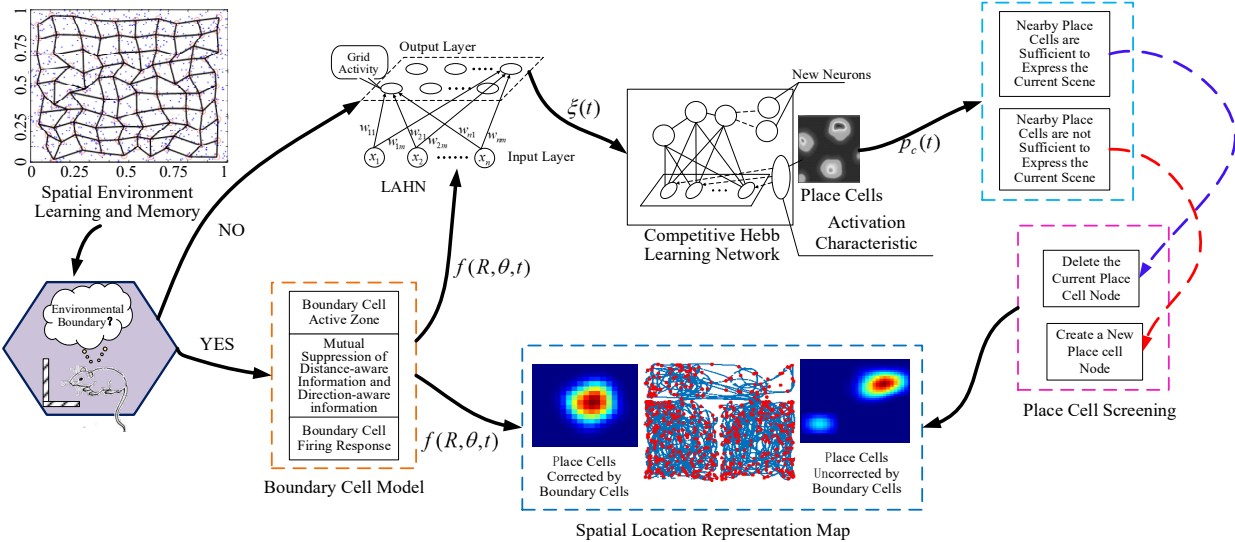

**Figure 6.** Schematic diagram of the spatial location representation map construction.

Step 1: As the mobile robot explores the environment, it collects the speed, direction, and distance-aware information needed to construct a spatial location representation map.

Step 2: The inputs of the direction and velocity-aware information to the input layer of LAHN are conducted. The output of the grid cell response value is obtained after the learning performed by LAHN.

Step 3: Grid cells undergo a competitive Hebb learning network in order to generate place cells that are capable of representing current location information [18]. The mobile robot explores the environment by continuously activating new place cells in response to new location scenarios and jointly constructs a spatial location cell representation map until the robot stops running. The algorithm for the construction of the spatial location representation map is shown in detail in Algorithm 1.

In order to avoid the undesirable situation where the number of place cell activations for the same scene is too many due to the small place cell spacing in the operation of the mobile robot (thus resulting in the waste of system computational resources) or the undesirable situation where the place cell spacing is too large (thus resulting in the poor accuracy of position estimation), this paper introduces the place cell distance threshold $r_{th}$ in order to constrain the place cell activation response. Figure 7 shows the box plot of the localization accuracy of the mobile robot when constructing the spatial location representation map under different values of $r_{th}$. The localization error in Figure 7 fluctuates upward with the distance threshold, and the localization error is minimized when $r_{th}$ is taken as 0.06 m, such that $r_{th}$ is taken as 0.06 $m$ in this paper.

---

**Algorithm 1:** Spatial location representation map construction algorithm

---

**Input:** Grid cell response value, place cell distance threshold $r_{th}$
**Output:** Spatial Location Representation Map
BEGIN:
FOR
Get grid cell response values
Updating winning place cells through competitive Hebb learning network
Calculate the Euclidean distance $r_b$ between Current place cell and nearby place cell
    IF $r_b < r_{th}$
        The previous place cell can represent the current scene, continue run forward
        ELSE
        The previous palace cell is not enough to represent the current scene and construct a new place cell
    END IF
    IF the movement is not over
        Continue forward motion and update grid cell response value information
        ELSE
Output spatial location representation map
END IF
ND FOR

---

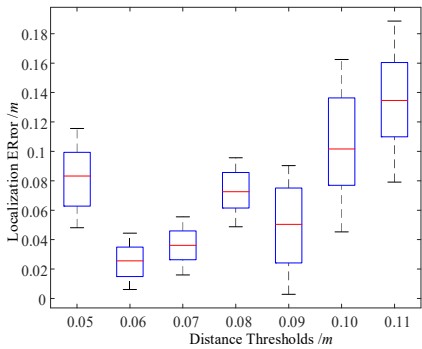

**Figure 7.** Comparison of the localization errors under different distance thresholds.

Step 4: When the mobile robot runs to the boundary cell excitation zone, the distance-aware information and angle-aware information of the mobile robot are relative to the environment boundary. They are mutually excited and inhibited in order to activate the boundary cells, which are used to correct the place cell response values and to eliminate the accumulated errors that are generated due to the long time running of the mobile robot. At the same time, the boundary cell response values are mapped to the input layer of LAHN, and the grid cell distribution law in the current scene is updated simultaneously. The place cell correction update rules are as follows:

$$p_{bc}(t) = \left[ p_c(t) + \left( \prod_{i=1}^{n} f_i(R, \theta, t) / \max_x \right)^{\frac{1}{n}} \right]_+ \tag{15}$$

where $f_i(R, \theta, t)$ is the boundary cell firing response value, $p_{bc}(t)$ is the place cell response value after the boundary cell response correction, and $p_c(t)$ is the place cell response value before the boundary cell response correction. $n$ represents the number of boundary cells.$[\bullet]_+$ indicates that the output value is non-negative.

Figure 8a depicts the pre-correction place cell response map, revealing the presence of an accumulated error resulting from prolonged operation of the mobile robot. This error is evident in the place cell response map, where a single place cell fails to generate a response to the current position. In Figure 8b, the post-correction position cell response map is presented. The comparison with Figure 8a demonstrates that the corrected place cell response map exhibits a solitary, distinct place cell response point, thereby enhancing the accuracy of current position estimation by eliminating interference caused by redundant place cell responses.

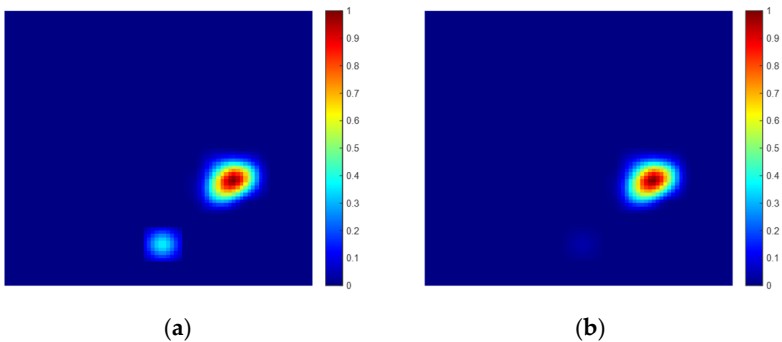

(a)            (b)

**Figure 8.** Place cell response maps. (**a**) Place cells corrected by boundary cells. (**b**) Place cells uncorrected by boundary cells.

## 4. Experimental Results and Analysis

The computer configuration used to test the experiments in this paper was as follows: i5-9400F CPU, 6-core processor, 2.9 GHz, 8 GB RAM. The method proposed in this paper was verified by the circular experimental datasets that were published in the Microstructure of a spatial map in the entorhinal cortex, published by Hafting et al. in Nature [11]. These datasets record the perceptual information, such as the movement direction, as well as the speed and distance of the rats at different times.

### 4.1. Boundary Cell Simulation Experiments

This paper further validates the method of this study using a larger Hafting circular experiment environment. The diameter of the circular experiment environment is 2 m and the rat also starts from the center of the experiment environment for the purposes of free exploration learning. The environmental plan diagram and the trajectory formed by the rat in completing the exploration are shown in Figure 9.

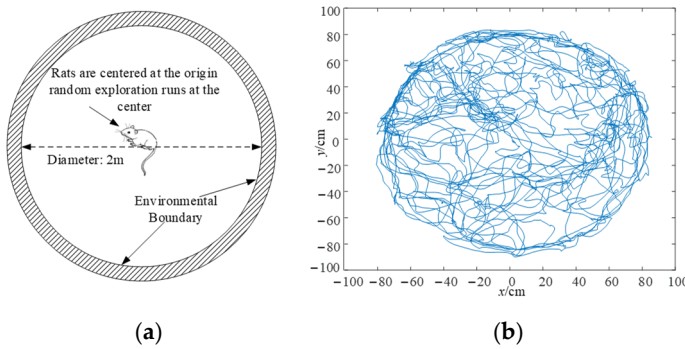

(**a**)                                    (**b**)

**Figure 9.** The environmental plan diagram and rat trajectory map. (**a**) The environmental plan diagram. (**b**) The rat trajectory map.

Figure 10 shows the intercepted boundary cell discharge response plots at different times. It can be seen that as the rat explores the environmental boundary gradually and comprehensively, the number of activated boundary cells in response to the environmental boundary increases. Table 2 shows the correlation data between the number of activated boundary cells and the mean localization error at different times. It can be seen that the number of activated boundary cells tends to increase rapidly before 1600 s, while after 1600 s, the number of activated boundary cells gradually slows down and stabilizes as the rat explores the environment more fully. At 1800 s, the activated boundary cells can adequately represent the environmental boundary and its number remains largely unchanged while the localization error is stable at about 0.037 m. It can be seen that the proposed method in this paper is also well adapted to larger circular environments.

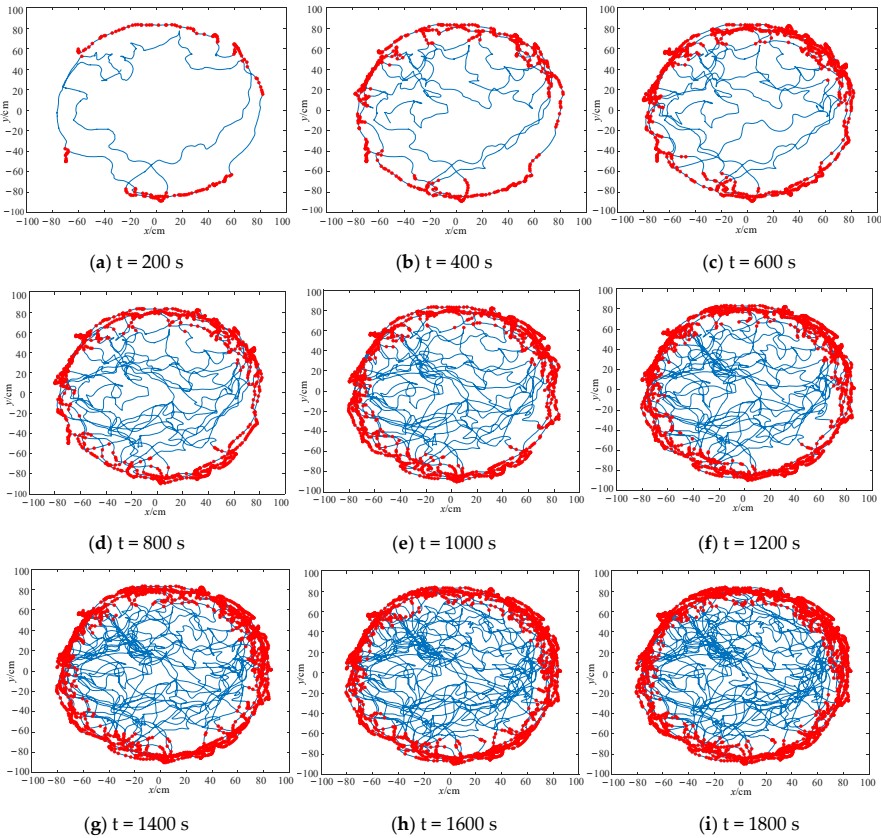

**Figure 10.** The boundary cell response maps at different times.

**Table 2.** The number of activated boundary cells at different times with the mean localization error.

| Time (s) | 200 | 400 | 600 | 800 | 1000 | 1200 | 1400 | 1600 | 1800 |
|---|---|---|---|---|---|---|---|---|---|
| Number of boundary cells (pcs) | 103 | 195 | 274 | 318 | 378 | 421 | 472 | 503 | 498 |
| Mean localization error (m) | 0.67 | 0.71 | 0.83 | 0.86 | 0.74 | 0.61 | 0.50 | 0.39 | 0.37 |

### 4.2. Grid Cell Simulation Experiment

The grid cell construction method proposed in this paper was validated by the Hafting circular experimental environment. Figure 11 shows the grid cell response maps obtained by the OI model, the CAN model, and the SLRB method at different times. The distribution of grid cells activated by the OI model after 30 min lacked physiological properties. In comparison, the grid cell distribution acquired by the CAN model has been improved, but the acquired grid cell clusters contain too many grid cells, posing a potential risk of computational time consumption for the construction of large-scale spatial location representation maps. Compared with the former two, the method in this paper obtained the vector information between the rat and the environment boundary, which was obtained by the rat in the process of exploring the environment and was performed to correct the grid cell distribution pattern. This meant that the method in this paper successfully achieved the goal of representing a circular experimental environment with fewer grid cells while maintaining the physiological characteristics of grid cells. Table 3 shows the number of activated grid cells and their corresponding grid cell fractions for the three compared methods in characterizing the above circular experimental setting. It is more intuitive to see from the data comparison of the three algorithms that the number of grid cells utilized by this method is lower. This was achieved under the premise of also achieving the purpose of characterizing the environment. Moreover, the fraction of the grid cells generated by this method is higher as the environment is gradually explored completely, which indicates that the grid cells activated by this method are highly active and reasonably distributed.

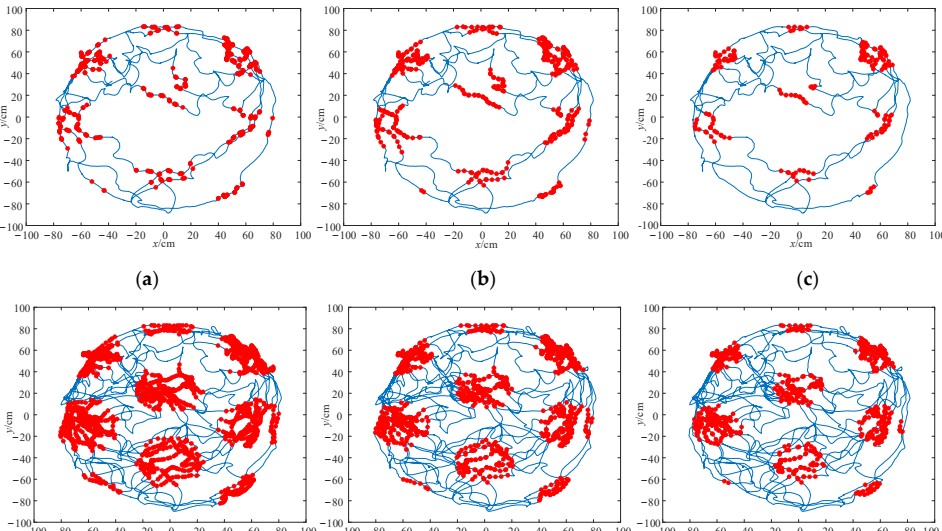

**Figure 11.** *Cont.*

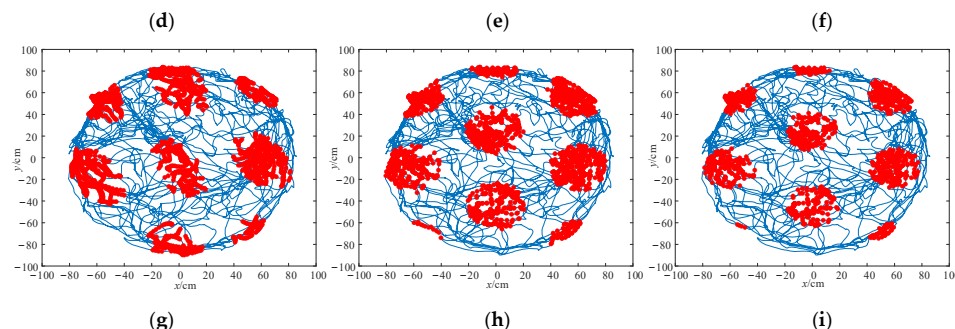

**Figure 11.** The grid cell response maps obtained by the OI and CAN models, as well as those obtained by the method used in this paper at different times. (**a**) 5 min for the OI model to obtain results. (**b**) 5 min for the CAN model to obtain results. (**c**) 5 min for this paper's method to obtain results. (**d**) 15 min for the OI model to obtain results. (**e**) 15 min for the CAN model to obtain results. (**f**) 15 min for this paper's method to obtain results. (**g**) 30 min for the OI model to obtain results. (**h**) 30 min for the CAN model to obtain results. (**i**) 30 min for this paper's method to obtain results.

**Table 3.** Comparison of the grid cell properties that were activated by the three algorithms.

| Exploration Time (min) | Number of Activated Grid Cells (pcs) | | | Grid Cell Fraction | | |
|---|---|---|---|---|---|---|
| | OI | CAN | SLRB | OI | CAN | SLRB |
| 5 | 171 | 160 | 128 | 0.71 | 0.75 | 0.74 |
| 10 | 382 | 171 | 165 | 0.73 | 0.74 | 0.69 |
| 15 | 410 | 290 | 195 | 0.68 | 0.79 | 0.79 |
| 20 | 472 | 353 | 287 | 0.54 | 0.71 | 0.84 |
| 25 | 524 | 427 | 354 | 0.51 | 0.72 | 0.88 |
| 30 | 614 | 541 | 478 | 0.43 | 0.65 | 0.86 |

*4.3. Spatial Location Representation Map Construction Experiment*

The place cell properties obtained by the method in this paper were validated in a Hafting circular experimental environment. Figure 12 shows the firing response maps of the pose/place cells to the current location as acquired by RatSLAM, as well as by the competing Hebb learning networks and the method used in this paper at different times. In addition, the maps of their spatial location representations are generated after completing the environmental exploration. The RatSLAM method still did not perform well in the circular environment because the RatSLAM algorithm only integrated the rat's self-motion information to activate the pose cells in response to the rat's current location, without exhaustively considering the effect of the simple similarity boundaries on the firing pattern of the pose cells. At the same time, along with the increase in the environmental scene, the growth of the exploration time response leads to a gradual increase in the accumulated error, which renders the RatSLAM method unable to generate a single pose cell by which to respond accurately to the current location. Compared with the RatSLAM method, the competitive Hebb learning network can adjust the response values of the place cells via the connection weights between the place cells, which enables the system to maintain a better localization performance at the early stage of unknown environment explorations. However, as the exploration proceeds and the environment itself is characterized, the number of place cells gradually increases and the burden of adjusting the connection weights between the place cells increases, thus leading to the overlapping phenomenon of place cell responses, which then affects the accuracy of localization. In order to overcome the negative effects of environmental size and cumulative errors, this method introduces environmental boundary information to correct the place cell firing responses in real time, such that only one place cell responds to the current location at each individual time.

By removing the interference of the other place cell firing responses, the accuracy of the location information estimation of this method is improved.

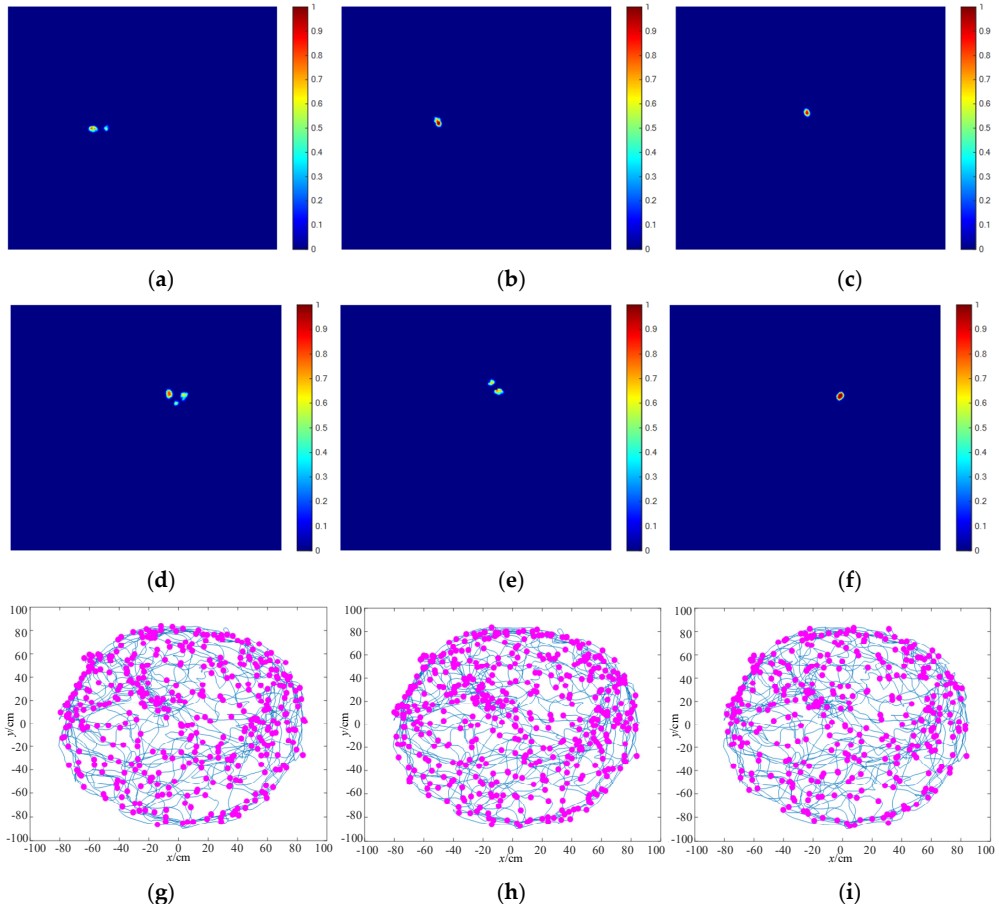

**Figure 12.** The process of constructing the spatial location representation maps by the three methods. (**a**) 5 min for the RatSLAM method to obtain results. (**b**) 5 min for the competitive Hebb learning network to obtain results. (**c**) 5 min for this paper's method to obtain results. (**d**) 30 min for the RatSLAM method to obtain results. (**e**) 30 min for the competitive Hebb learning network to obtain results. (**f**) 30 min for this paper's method to obtain results. (**g**) The RatSLAM spatial location representation map. (**h**) The competitive Hebb Learning network spatial location representation map. (**i**) The SLRB method spatial location representation map.

Figure 13 shows a comparison of the experimental data that was generated by the three methods for constructing spatial location representation maps. From Figure 13a, it can be seen that the number of place cells activated by all three methods in the early stage of the environmental information exploration grew rapidly with time, but with the completion of the environmental learning in the rats, the growth of grid cells in this method entered a stable interval after 20 min. Furthermore, the final number was stabilized at about 300, which was reduced by about half when compared with the other two methods. In terms of the absolute trajectory error shown in Figure 13b, the absolute trajectory error of the method in this paper at the end of the run is about 0.031 m, which is about 47.2% lower when compared to the competitive Hebb learning network and about 56.9% lower when compared to the RatSLAM algorithm. Additionally, in terms of the whole exploring time, the method in this paper shows a good performance in terms of the location estimation.

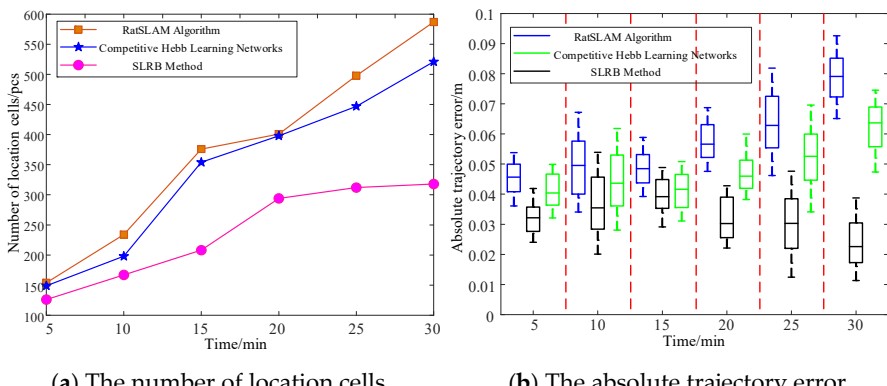

(**a**) The number of location cells

(**b**) The absolute trajectory error

**Figure 13.** Comparison of the performance of the three methods in terms of their spatial location representation maps.

## 5. Analysis and Discussion

Discussion 1: Experiments on Activated Boundary Cells in Different Scenarios

In the experiments involving activated boundary cells in different scenarios, as the rat runs facing a wall, the number of activated boundary cells gradually increases, transitioning from the inhibition zone of boundary cells to the region of increased activity. When the rat moves away from the environmental boundary, the activation frequency of boundary cells gradually decreases (with a slower growth rate in the number of activated boundary cells). When the rat's motion corresponds to Figure 3d, exhibiting a trend of moving away from the environmental corner, the activation frequency of boundary cells gradually decreases. In an instantaneous motion similar to Figure 3d but with the difference that the rat is moving away from the environmental corner, and the restraining force from the environment decreases rapidly as the rat moves further from the boundary, the number of activated boundary cells temporarily enters a stage of slow growth. It can be observed that the activated boundary cells in different scenarios exhibit similar patterns as shown in Figure 4 (Activated Boundary Cells in Different Scenarios), aligning with the physiological observations. Furthermore, from the boundary cell response graphs at different time points in Figure 10, it can be observed that the proposed model can effectively model the boundary cells based on boundary information, regardless of the square or circular environment. As indicated by the boundary cell discharge response graphs captured at different time points in Figure 10, with the rat's comprehensive exploration of the environmental boundary, the number of activated boundary cells responding to the boundary continuously increases. Before 1600 s, the number of activated boundary cells exhibits a rapid increase, while after 1600 s, with the rat's comprehensive exploration of the environment, the growth rate of the number of activated boundary cells gradually slows down, approaching stability. At 1800 s, the activated boundary cells sufficiently represent the environmental boundary, and the number of activated boundary cells remains relatively stable, while the localization error stabilizes at around 0.37 m. This demonstrates the adaptability of the proposed method to larger circular environments, validating the effectiveness of the algorithm presented in this paper.

Discussion 2: Validation Experiment on the Public Hafting Dataset

The method proposed in this paper is validated using the square experiment dataset and the circular experiment dataset published by Hafting et al. in their paper "Microstructure of a spatial map in the entorhinal cortex" in Nature. The obtained position cell responses from the method proposed in this paper are compared with the position cell responses obtained from the competitive Hebbian learning network and the pose cell discharge response map obtained from RatSLAM. The RatSLAM algorithm only integrates self-motion information from the rat to activate pose cell responses at its current location, without fully considering the influence of simple geometric boundary cues on the discharge patterns of pose cells. Additionally, with the increase in exploration time response and

the growing environmental scene, cumulative errors gradually accumulate, resulting in RatSLAM's inability to generate a single pose cell that accurately responds to the current location. Compared to RatSLAM, the competitive Hebbian learning network can adjust the response values of position cells through the connection weights between them, enabling the system to maintain good localization performance in the early stages of exploration in unknown environments. However, as exploration progresses and due to the characteristics of the overall environment, the number of position cells increases, and the burden of adjusting connection weights between position cells increases, leading to overlap in their response patterns and consequently affecting the accuracy of localization. The proposed method in this paper activates boundary cells when encountering boundaries, and the correction function of boundary cells on position cells enables real-time updating and constraint of position cell discharge responses, reducing position estimation uncertainty, and thereby improving the localization accuracy of the proposed method. At the end of the operation, the absolute trajectory error of the proposed method is approximately 0.031m, which is about 47.2% lower than that of the competitive Hebbian learning network, and about 56.9% lower than that of the RatSLAM algorithm.

## 6. Conclusions

In this paper, based on the understanding of the physiological properties of various spatial navigation cells and their role in autonomous navigation and localization, a spatial location representation method incorporating boundary information is proposed in order to construct a map of the unknown environment. The method improves the accuracy of autonomous localization and the robustness of map construction by activating the learning and memory of the spatial location of the unknown environment by navigation cells. The method presented in this paper belongs to an exploration of the mechanism of the brain operations that occur during the mammalian process of localization and map construction. It lays the foundation for further research on bionic localization and navigation algorithms for mobile robots. However, the method proposed in this paper only utilizes self-motion information such as rat's direction, velocity, and distance for mapping and does not consider the influence of visual perceptual information on mapping. This limitation results in the inability of the method to perform relocalization using familiar scenes. When fusing visual information with self-motion information, the difference in the sampling rates of the two signals can lead to joint initialization failure. Future work will propose a joint initialization method for visual and self-motion information to synchronize the two signals and overcome the challenges of joint initialization. By incorporating the obtained visual perceptual information into the proposed method, the stability and accuracy of spatial representation map construction will be improved.

**Author Contributions:** Conceptualization, H.J.; Funding acquisition, H.J.; Software, H.J.; Validation, Y.Z.; Writing—original draft, H.J.; Writing—review and editing, Y.Z. All authors have read and agreed to the published version of the manuscript.

**Funding:** This work is supported by Robotics Control Technology Research Center of Huainan Union University (LZX2201), and The cooperation R&D project of Anhui Sound Valley Intelligent Technology Co.

**Institutional Review Board Statement:** Not applicable.

**Informed Consent Statement:** Not applicable.

**Data Availability Statement:** The data related to this paper are available on request from the corresponding author.

**Conflicts of Interest:** The authors declare no conflict of interest.

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
