# Peer review of "A Spatial Location Representation Method Incorporating Boundary Information"

_applsci, doi:10.3390/app13137929_

Round 1
Reviewer 1 Report
- The authors proposed the SLRB method that needs less activated cells to achieve better results than the OI and CAN methods.
- This is a very interesting paper, very well written and also very well structured.
- In equation 1, what does b mean? There are some nomenclature that is not defined. There are variables in the equations where the author does not specify their meaning or what they refer to.
- Many mathematical equations appear in the paper without any explanation, and in some cases the equations lack a reference (to its origin).
- Figure 7, How did the authors reach the value of 0,06m for r_th? Is this the best value? The authors rely only on the results obtained from figure 7?
- Table 2, Why does the mean localization error increase between 200 and 800 (s) e after 1000 (s) starts to decrease?
- At subsection 4.2. the authors do not define the acronym CAN. Nevertheless it can be considered common sense, it must be defined in the paper.
- It would be important to verify the results in a real robotic platform, so as to verify if the results are the same as the ones in simulation.
- The paper is written in good English.
Author Response
请参阅附件。

Reviewer 2 Report
The authors propose a spatial location representation method incorporating boundary information (SLRB). This method is inspired by the spatial cognitive mechanism in mammals. They aim to understand rat movement and generate movement boundaries for spatial location representation. They compare their proposed method with different algorithms and find that it requires fewer location cells and results in less trajectory error. However, the testing of their algorithm was conducted only in simulation, so it is necessary to test it using a mobile robot. Overall, the paper is well-written.
Regarding your specific questions about the figures:
- Figure 1: It would be helpful if there were more explanations and symbols provided in this figure. It seems that an explanation of the environmental characterization map is missing.
- Figure 3: It is unclear which criteria were selected in this figure. It would be beneficial to provide more information about whether it covers all unknown environments.
The reviewer suggests that in Figure 4, it would be beneficial for the authors to provide a clear presentation of the boundary cell activity inhibition zone, the attenuation zone, and the growth zone.
- "fi( R , angle , t )" is not explained in the context you provided.
- Figure 8: The explanation of this figure does not match the sentence provided. It is unclear what the correct explanation should be.
Overall, the paper is well-written.
Reviewer 3 Report
The paper shows a spatial location representation method (SLRB). It considers the low autonomous localization accuracy and cumulative errors in mobile robots navigating unknown environments. In this paper, the proposed method reduces localization errors and corrects accumulated errors by using boundary information and utilizing self-motion data.
Experimental results prove an improvement in absolute trajectory error wtih respect to the available ones.
1- The authors should clearly present the specific problem of low autonomous localization accuracy and cumulative errors faced by mobile robots in unknown environments in the abstract.
2- The authors should present the main contribution also in the abstract.
3- In the introduction, recent references should be added.
4- The authors should clearly discuss the relationship between the experimental scenarios in Figure 3 and the boundary cell activation response maps in Figure 4.
5- The authors should clearly discuss the effects of the behavior of the rats in different scenarios.
6- The authors should clearly discuss the variations in boundary cell activation frequency according to movement.
7- The authors should clearly discuss the superior sides of the method leading to a reduction in the absolute trajectory error with respect to the Hebb learning and RatSLAM.
8- The authors should discuss challenges and potential solutions for adding visual perception information in the conclusion also.
Reviewer 4 Report
1. Comparing Figure 2 and Figure 3, it seems that there is a problem with the parameter definition in Figure 2. For example, shouldn't the starting point of d_r be the center of the robot? The figure needs to be edited to be consistent. Also, what is ‘b’ and ‘r’? all the parameters are defined before they are used.
2. Lack of explanation for each scenario presented in Figure 3 makes the reader's understanding of this thesis low. Add clear descriptions of the characters used in each scenario and picture. What is the difference between r1 and r2 and in some cases r1 is on a vertical line while in other cases it is on a horizontal line, what difference does it make?
3. This study introduced an algorithm that applies spatial recognition mechanisms of mammals (rats). The environments used were triangular, pentagonal, pentagonal, and circular. The localization performance improvement through the analysis of the activated cluster is confirmed, but an additional explanation of the part that improves the location accuracy for a specific location in localization is required. For example, in the case of a general circular environment, there are no feature points for each position, and it is necessary to explain by which mechanism the positional accuracy of the robot is improved.
It is good.
Round 2
Reviewer 4 Report
All comments were answered and the relevant parts were applied to the text and corrected. It is available for publication in its current state.